# Performance of Fiber-Reinforced Alkali-Activated Mortar with/without Nano Silica and Nano Alumina

**Mahmood Hunar Dheyaaldin** [1,*] **, Mohammad Ali Mosaberpanah** [1] **and Radhwan Alzeebaree** [2,3]

1   Civil Engineering Department, Cyprus International University, Nicosia 99010, North Cyprus, Turkey; mosaberpanah@gmail.com
2   Akre Technical Institute, Duhok Polytechnic University, Duhok 42004, Iraq; radhwan.alzeebaree@dpu.edu.krd
3   Civil Engineering Department, Nawroz University, Duhok 42002, Iraq
*   Correspondence: mahmood.dheyaaldin@gmail.com; Tel.: +964-7504244554

**Abstract:** The current study is aimed to evaluate the effect of nanomaterials (nano alumina (NA) and nano silica (NS) on the mechanical and durability performance of fiber-reinforced alkali-activated mortars (FRAAM). Polypropylene fiber (PPF) was added to the binders at 0.5% and 1% of the volume of the alkali-activated mortar (AAM). Design-expert software was used to provide the central composite design (CCD) for mix proportions. This method categorizes variables into three stages. The number of mixes was created and evaluated with varied proportions of variables. The primary binders in this experiment were 50% fly ash (FA) and 50% ground granulated blast slag (GGBS). The alkali-activated solution to binder ratio was 0.5, and the sodium hydroxide (NaOH) concentration was 12 molarity. The sodium silicate to sodium hydroxide ratio was 2.5. The cubic specimens and prisms were evaluated in an ambient atmosphere at 23 + 3 °C room temperature at the ages of 7 and 28 days. The mechanical performance of AAM was indicated through evaluation of the compressive and flexural strength, flowability, and unit weight of the alkali activator mortar. In addition, the durability performance and microstructure analysis were also evaluated. The experiments demonstrated that the AAM without fibers and nanomaterials had a higher flow rate than the other mixtures. However, the flowability of all mixtures was acceptable. The highest compressive strength was deducted through the use of 2% NA and higher flexural tensile strength was obtained for mixtures included 1% NS and 0.5% PPF. The lower water absorption was noted through the combination of 2% nano silica and 1% polypropylene fiber. Whereas, the combination of 2% nano silica, 1% nano alumina, and 0.5% polypropylene fiber had the lower sorptivity. In addition, the microstructure analysis indicated that the nanomaterials significantly improved the matrix and the porosity of the matrix was considerably reduced.

**Keywords:** alkali-activated mortar; nano alumina; nano silica; polypropylene fiber; mechanical properties; durability; sorptivity; microstructure analysis

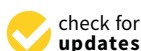



## 1. Introduction

In general, alkali-activated concrete is one of the inorganic polymers. It is more amorphous than crystalline compared with other natural zeolitic minerals [1]. Polymerization in alkaline conditions requires a substantially faster reaction of silica (Si) alumina (Al), resulting in a three-dimensional polymeric chain of poly(sialate) Si–O–Al–O connections. Alkali-activated concrete is produced using traditional Portland cement (OPC) or pozzolanic cement. Alkali-activated concrete has a high alkali concentration as well as a wide range of silica (Si) and alumina (Al) content [2]. Geopolymer concrete has a high alkali concentration and a wide range of silica (Si) and alumina (Al) content compared with regular conventional Portland cement (OPC) or pozzolanic cement. Materials used for alkali-activated concrete production with high amounts of silica and alumina, such as fly ash (FA), ground granulated blast slag (GGBS), metakaolin (MK), and rice husk (RHA),

are examples of alumino-silicate-based alkali-activated concretes. RHA is a green product that is generally utilized to generate power or as boiler fuel for rice processing. It is an indigestible outer husk that is removed and burned to create steam for boiling rice, either in household stoves or in local power plants [3,4]. The replacement ratios of RHA used were 20% to 25% of the weight of the binder as ash for producing power or as boiler fuel for processing paddy [3,4]. Furthermore, the silica in pozzolana mixes with the portlandite produced during the hydration of ordinary Portland cement (OPC), contributing to the development of its strength [5]. Silica fume (SF) or other essential additive producers can provide specifications for silica fume concrete with high durability or strength [6,7]. The environmentally responsible disposal of these waste materials necessitates appropriate techniques to reduce greenhouse gas emissions into the atmosphere. The world's Earth Summits urged the cement industry to transition away from Portland cement and toward a more environmentally friendly alternative binder with desirable structural and durability properties [8]. The type of alumina-silicate-based material used to make alkali-activated concrete (GPC) is determined by the material's cost and availability, as well as the application [9]. Alkali-activated concrete is frequently produced from industrial byproducts such as fly ash and slag. Fly ash is produced as a byproduct of the combustion of pulverized coal. Fly ash has been created in large quantities due to industrialization, and it has accumulated over time. According to surveys, the world produces about 780 million tons of fly ash each year, but only 17–20% is used. More than 220 million tons of fly ash are produced in India each year, but only 35–50% is used [10,11]. Several studies studied the production of alkali-activated concrete and mortar using fly ash (FA) and ground granulated blast slag (GGBS). The production of alkali-activated concretes significantly reduces the industrial waste by at least 12.2 million tons per year while emitting five to six times less $CO_2$ [12,13]. Synergic usage of GGBS and fly ash revealed more promising behavior than fly-ash-based alkali-activated concrete. The mineral composition of the material is directly related to the development of strength. GGBS and fly ash include considerable amounts of $SiO_2$, $CaO$, and $Al_2O$ [14]. Most research has focused on the mechanical and microstructural characteristics for alkali-activated concretes, and how different additions affect these properties.

Nanomaterial and/or nanopowders with various materials offer significant benefits over other additives, including superior mechanical characteristics and long-term durability for alkali-activated composites. Their high specific area is especially notable [15]. The addition of NS increased the compressive strength for both the lime–pozzolan and the lime–metakaolin composites. Furthermore, NS reduced carbonation and water absorption in both composites and provided denser microstructures. The addition of nano alumina to pozzolanic binders produced a novel behavior. Except for the volume change of the specimens, alumina nanoparticles significantly enhanced all of the system's properties. The development of C-A-S-H molecules can adequately explain this phenomenon. Alumina nanoparticles not only functioned as a filler to keep the microstructure together, but they also appear to have a positive influence on the evolution of these compounds [16]. The self-aggregation of nanoparticles reduces the small size advantages of nanomaterials and produces unreacted pockets that contribute to stress development during concrete production [17]. Using nanoparticles in construction offers multiple advantages that must be addressed due to their structural, environmental, and economic benefits, and many practical studies for the use of various types of nanomaterials are available. Despite their numerous benefits, many of these materials are low cost to implement in essential building phases and are inexpensive to purchase because they are cheap materials. Thus, these materials might have specific disadvantages, for example, the restricted usage of nanoparticles is owing to a lack of research revealing the long-term negative effects of these materials on public health, as well as a lack of technical standards used in their production and development [18]. Alkali-activated concrete (AAC) can be produced without the use of Portland cement (PC), making it a green or eco-friendly concrete. In addition to decreasing $CO_2$ emissions, alkali-activated concrete (AAC) has been demonstrated to have high mechanical

characteristics. Furthermore, alkali-activated concrete (AAC) produces 44–70% less $CO_2$ than conventional concrete [19,20]. Geopolymerization is a chemical reaction that occurs in an inhomogeneous form between alkali solutions and silicate-alumino oxides. In highly alkaline settings, squishy, patchy to semi-crystalline composites with Si–O–Si and Si–O–Al bridges form. A Si- and Al-rich source material mixes with a highly alkaline solution to generate a binding material that simulates a network of poly(sialates) and is amorphous to semi-crystalline in nature [12,21]. Several researchers have attempted to replace Portland cement (PC) with a more environmentally friendly concrete that incorporates various byproduct elements, which are further discussed in the following subsections. Because of its critical role in lowering the quantity of pollutants and $CO_2$ emissions created during Portland cement (PC) production, alkali-activated concrete has shown a paradigm change in building industries throughout the world [12]. Melted slag contains around 40% calcium oxide (CaO) and 30–40% silicon dioxide ($SiO_2$), which is similar to the chemical production of conventional Portland cement (PC). All binders examined for replacement rates of up to 40% wt had consistently high compressive strengths (about 60 MPa for metakaolin-based binders and 52 MPa for fly-ash-based binders) [22]. The binding procedures and materials utilized in concrete production are also regarded as essential [12]. Concrete's long-term strength, sulfate and alkali-silica reaction resistance, and water demand, permeability, and heat output can all be improved using ground granulate blast slag (GGBS) [23,24]. In certain countries, fly ash (FA) can cost 20% to 60% less than regular Portland cement (PC), but in others, PC can be more than twice as expensive as fly ash [25]. Fly ash, on the other hand, is rarely imported over great distances and is more expensive than local OPC since specific concrete durability standards can only be met by utilizing fly ash. This material can also help the environment by conserving landfill space, decreasing water and energy use, and lowering greenhouse gas emissions [26].

The alkaline solution includes both alkali silicates and hydroxides, used without the presence of distilled water, that have improved in terms of durability and resistance to external attacks [27]. The strength of alkali-activated mortar (GPM) is controlled by components such as calcium content, sodium hydroxide (NaOH) molarity, and the binder/aggregate, solution/binder, $Na_2SiO_3$/NaOH, silicate/$Na_2O$, and silicate/Al ratios. Furthermore, the source material, curing conditions, and particle-size distribution all have a significant influence on the development of the compressive strength of alkali-activated mortar [28]. In typical construction practice, concrete elements are left in average temperatures after casting instead of curing at high temperatures. When slag is used as a partial replacement for fly ash (FA) or metakaolin in source materials, the alkali-activated concrete can set at a constant temperature and gain greater strength at early and long-term ages [29,30]. As a partial replacement for ground granulated blast slag (GGBS), mixes including up to 50% fly ash can reduce disposal costs as well as the environmental effect of such byproducts [31]. As a result, carbon dioxide emissions from cement manufacturing may be decreased, as can the building industry's high energy and natural resource consumption, contributing to enhanced development and sustainability [31]. On the other hand, an excess of salt in the solution may cause sodium carbonate to form as a result of air carbonation. As a result, for high-strength fly-ash-based alkali-activated concrete, a sodium silicate/sodium hydroxide ratio of 1.5 to 2.5 is recommended [32].

The poor tensile strength and ductility of alkali-activated concretes led to the utilization of fibers to avoid this weakness [33]. The use of fibers increased the concrete's post-cracking, ductility, and toughness [34,35]. Furthermore, fiber-reinforced alkali-activated concrete has a better cost/benefit ratio than concrete without fiber [36]. The amount and aspect ratio of fibers is a major consideration in the design approach and optimization technique used to produce concrete mixtures [37–39]. Researchers considered 1% to be an appropriate quantity of fibers for the majority of structures and economic concerns [40]. The load was applied until the fibers separated from the matrix. This approach allows for more energy absorption, which results in a more stable fracture process with increased fracture energy. When a sufficient quantity of polypropylene fiber is dispersed throughout the matrix to

bridge any emerging micro fractures, breaking or pulling out the fibers needs more energy, increasing the material's failure load and toughness. Polypropylene fibers also have a significant influence on the flexural strength of alkali-activated concrete. Flexural strength was 7.03, 7.51, and 7.69 for polypropylene fibers with monomer ratios of 2%, 2.5%, 3%, and 0.3%, respectively [41].

Although there are several studies regarding the use of nanomaterials and polypropylene fiber in ordinary concretes, few investigations studied the combined effect of nanomaterials and polypropylene fiber on the performance of alkali-activated mortar. In the current study, the performance of alkali-activated mortar including nanomaterials (nano silica (NS) and nano alumina (NA)) with different percentages of polypropylene fiber was studied. The influence factors were divided into three-level variables and analyzed by central composite design (CCD) to generate an optimum mix design of a fly ash (FA)/ground granulated blast slag (GGBS)-based alkali-activated mortar. In addition, durability and microstructure analyses were also conducted to clarify the effect of nanomaterials and polypropylene fibers in detail.

## 2. Materials

To examine the influences of nano silica (NS), nano alumina (NA), and polypropylene fiber (PPF) on the freshness, mechanical, durability, and microstructure properties of alkali-activated mortar, 15 variants of alkali-activated mortars (AAM) were prepared, with and without NS, NA. The nanomaterials ratios were (0%, 1%, and 2%) by the weight of binders material and the PPF ratios were (0%, 0.5%, and 1%) by the volume of AAM. Locally available class F fly ash (FA) according to ASTM C 618 [42] and ground granulated blast slag (GGBS) were utilized as binder materials in the production of AAM. The chemical composition analyses of the powder materials was conducted using X-ray fluorescence (XRF) as shown in Table 1. Table 2 illustrates the physical properties of the binders and additives. Locally available natural sand was used as fine aggregates, and the sieve analysis for the fine aggregates is shown in Figure 1. A solution of sodium silicate ($Na_2SiO_3$) and sodium hydroxide (NaOH) was used as the alkali-activated solution. The sodium silicate was obtained from a local supplier ($Na_2O$:17.98%, $SiO_2$: 28.1, water: 54.12% by mass). Sodium hydroxide with 98% purity and flaky shape was used with a molarity of 12 M, which was found to be the optimum concentration in the production of AAM [43,44]. the polypropylene fibers (PPF) were 6 mm in length. The nano alumina (NA) and nano silica (NS) particle sizes were 20–30 nm [45–47]. Glenium 51 basf superplasticizer (SP) was used to attain suitable workable mixes.

**Table 1.** Chemical compositions of fly ash (FA), ground granulated blast slag (GGBS), nano alumina (NA), and nano silica (NS).

| Component | CaO | $SiO_2$ | $Al_2O_3$ | $Fe_2O_3$ | MgO | $TiO_2$ | $SO_3$ | $K_2O$ | $P_2O_3$ | $Mn_2O_3$ | $Na_2O$ | SrO | L.oI |
|---|---|---|---|---|---|---|---|---|---|---|---|---|---|
| FA% | 15.48 | 48.43 | 17.15 | 11.96 | 1.35 | 2.68 | 0.82 | 0.41 | 0.4 | 0.17 | 0.0019 | 0.2 | 1.47 |
| GGBS% | 47.75 | 28.17 | 8.6 | 0.42 | 3.89 | 0.94 | 1.45 | 0.29 | 0.06 | 0.47 | 0.02 | 0.076 | 0.2 |
| NA% | - | - | 99.9 | - | - | - | - | - | - | - | - | - | - |
| NS% | | 99.8 | | | | | | | | | | | |

**Table 2.** Physical properties of fly ash (FA), ground granulate blast slag (GGBS), and polypropylene fiber (PPF).

| Physical Properties | Specific Surface Area m²/kg | Size μm | Density g/cm³ | Moisture Content % | Colour | Length mm |
|---|---|---|---|---|---|---|
| GGBS | 418 | - | 2.9 | 0.1 | Light grey | - |
| FA | 360 | <45 | <2.6 | <1.0 | Grey | - |
| PPF | - | 13 | - | - | White | 6 mm |

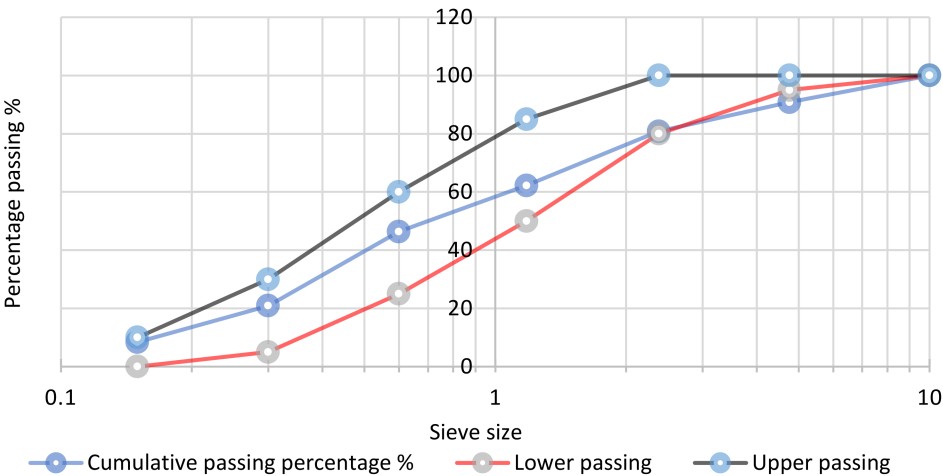

**Figure 1.** Sieve analysis for natural fine aggregate.

## 3. Mix Design and Methods

A constant total binder concentration of 700 kg/m$^3$ was used to create a series of alkali-activated mixtures. The mixtures contained 50% fly ash (FA) and 50% ground granulated blast slag (GGBS) by weight. Nano silica and Nano alumina were incorporated at 1% and 2% by the weight of the binder, respectively. Polypropylene fibers (PPF) were added to the mortar at 0.5% and 1% by volume. The weight of the mixture component in 1 m$^3$ AAM was shown in Table 3. The Design–Expert statistics computer software was used to carry out the mix design through the central composite design (CCD) method. The CCD approach was used to evaluate the individual and synergistic effects of three components with three levels on a particular response. This technique can decrease the number of experiments required to determine the major effect and interactions of each [48,49]. Table 4 presents the boundaries of the selected factors; polypropylene fiber (PPF), nano silica (NS), and nano alumina (NA). The percentage ratios of nano silica and nano alumina were (0%, 1%, and 2%) and the ratios of polypropylene fiber were (0%, 0.5%, and 1%). The signs before the numbers noted (−1: lowest, 0: middle, and 1: highest) the ratio of the current materials. The Na$_2$SiO$_3$/NaOH ratio was 2.5 [50]. The mixing procedure started with blending fine aggregates, fly ash, and ground granulated blast slag for 2.5 min. The alkali-activated solution and superplasticizer (SP) were added to the mixture and mixed for an additional two minutes. Finally, the fibers (for the mixes including PPF fibers) were added slowly and blended for an additional two minutes [50]. For each mix, three identical samples were cast for each test: the compressive and flexural strength tests. The average value was measured for each mix.

**Table 3.** Mix design and quantity of material per kg/m$^3$.

| Mix No. | NS | NA | PPF | FA | GGBS | S.H. | S.S. | F. Agg. | SP | E.W. |
|---------|----|----|-----|----|------|------|------|---------|----|------|
| M1 | −1 | 1 | −1 | 343 | 343 | 100 | 250 | 1033.5 | 21 | 33.35 |
| M2 | −1 | 1 | 1 | 343 | 343 | 100 | 250 | 1033.5 | 21 | 33.35 |
| M3 | −1 | −1 | 1 | 350 | 350 | 100 | 250 | 1033.5 | 21 | 33.35 |
| M4 | 1 | 1 | −1 | 336 | 336 | 100 | 250 | 1033.5 | 21 | 33.35 |
| M5 | 1 | 1 | 1 | 336 | 336 | 100 | 250 | 1033.5 | 21 | 33.35 |
| M6 | 1 | −1 | −1 | 343 | 343 | 100 | 250 | 1033.5 | 21 | 33.35 |
| M7 | −1 | −1 | −1 | 350 | 350 | 100 | 250 | 1033.5 | 21 | 33.35 |
| M8 | 1 | −1 | 1 | 343 | 343 | 100 | 250 | 1033.5 | 21 | 33.35 |

**Table 3.** *Cont.*

| Mix No. | NS | NA | PPF | FA | GGBS | S.H. | S.S. | F. Agg. | SP | E.W. |
|---------|----|----|-----|-----|------|------|------|---------|-----|------|
| M9 | 0 | 0 | −1 | 343 | 343 | 100 | 250 | 1033.5 | 21 | 33.35 |
| M10 | 0 | 1 | 0 | 339.5 | 339.5 | 100 | 250 | 1033.5 | 21 | 33.35 |
| M11 | 0 | 0 | 1 | 343 | 343 | 100 | 250 | 1033.5 | 21 | 33.35 |
| M12 | 0 | −1 | 0 | 346.5 | 346.5 | 100 | 250 | 1033.5 | 21 | 33.35 |
| M13 | −1 | 0 | 0 | 346.5 | 346.5 | 100 | 250 | 1033.5 | 21 | 33.35 |
| M14 | 1 | 0 | 0 | 339.5 | 339.5 | 100 | 250 | 1033.5 | 21 | 33.35 |
| M15 | 0 | 0 | 0 | 343 | 343 | 100 | 250 | 1033.5 | 21 | 33.35 |

Note: Abbreviations: NS—nano silica, NA—nano alumina, PPF—polypropylene fiber, FA—fly ash, GGBS—ground granulated blast slag, S.H.—sodium hydroxide, S.S.—sodium silicate, F. Agg.—fine aggregate, SP—superplasticizer, and E.W.—extra water.

**Table 4.** The variable ranges of additives.

| | The Variable Ranges | | | |
|---|---|---|---|---|
| **Additive Variables** | **Assigned** | **Level of Additive Variables** | | |
| | | **−1** | **0** | **1** |
| Nano silica | A [a] | 0.0% | 1.0% | 2.0% |
| Nano alumina | B [b] | 0.0% | 1.0% | 2.0% |
| Polypropylene fiber | C [c] | 0.0% | 0.5% | 1.0% |

[a] Friction of binder material addition. [b] Friction of binder material addition. [c] Friction of fiber addition.

## 4. Curing Method

Specimens were covered with a plastic sheet to prevent the alkaline solution from evaporating during the 24 h following the formation of the alkali-activated mortar. After 24 h, the samples were demolded and placed in a plastic bag in a laboratory room (23 ± 3 °C) according to ASTM C109 [51] until date of the hardened concrete test [52]. Three identical samples for each mixture were used for both the compressive strength [51] and the flexural tensile strength tests shown in Figure 2. The flexural strength of the samples was evaluated according to ASTM C348 standard [53].

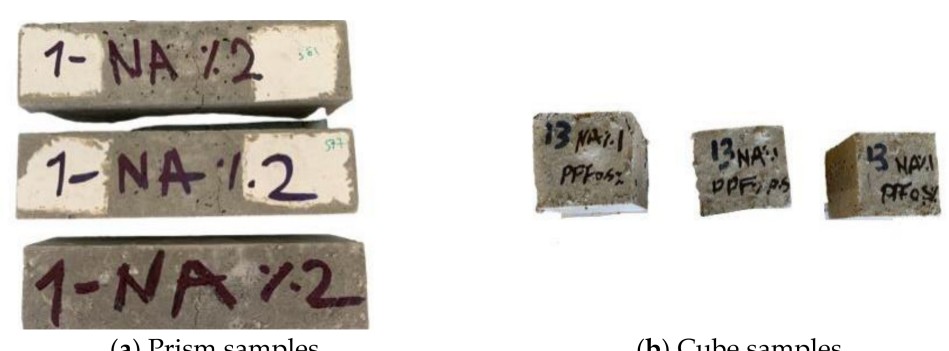

(**a**) Prism samples　　　　　　　　(**b**) Cube samples

**Figure 2.** The cubic and prism specimens.

## 5. Testing Procedure

### 5.1. Flow Table Test

The workability of the alkali-activated mortar was determined using the flow table test, ASTM C230 [54], which summarized the size of the cone: the diameter of the bottom is 100 mm, the top diameter is 70 mm, and the cone height is 50 mm. Figure 3 shows the flow table and the flow of alkali activated mortar. The mold was filled with two layers

of fresh alkali-activated mortar, which was tamped 20 times to ensure uniformity. A flow table instrument was used to place the cone. The mold was cleaned, lubricated, levelled, and then dropped 25 times in 15 s while the mold was immediately lifted vertically. The values of flow were an average of four symmetrical diameters on the table. High, moderate, or stiff workability was measured from the flow table values. The standard workability ranges and the diameter flow values for the mortar are shown in Table 5 [24].

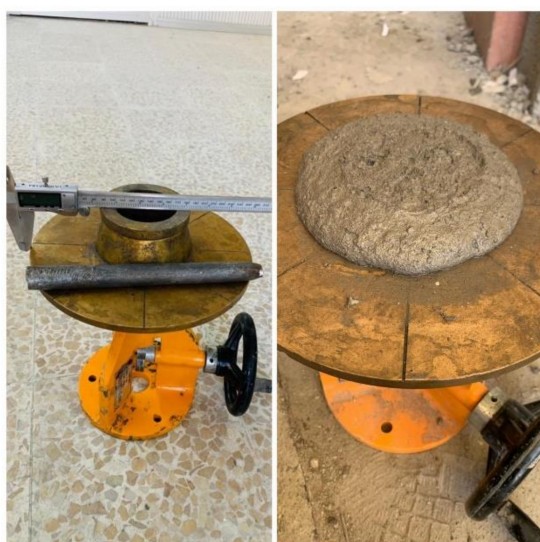

**Figure 3.** The flow table test.

**Table 5.** The flow value and workability range for alkali-activated mortar [55].

| No. | Diameter of Flow | Workability Range |
|---|---|---|
| 1 | Above 25 | (Very high) |
| 2 | 18–25 cm | (High) |
| 3 | 15–18 cm | (Moderate) |
| 4 | 12–15 cm | (Stiff) |
| 5 | Below 12 cm | (Very stiff) |

*5.2. Bulk Density*

The unit weight of AAM was evaluated according to the standard ASTM Cl38 [56]. The cylinder mold was used to conduct the unit weight test. First, the volume of the cylinder mold was determined. Then, the mold was filled with AAM and weighted. Finally, the empty mold and the mortar-filled mold were both weighed separately. To determine the weight per unit area, the following equation was used:

$$\text{Bulk density} = \frac{M_f - M_e}{V} \tag{1}$$

$M_f$ is the weight of the mortar-filled mold
$M_e$ is the empty mold's weight
$V$ is the mold's volume

*5.3. Water Absorptions*

Permeability is the most critical determinant of concrete and mortar for long-term performance. The ability of components to enter and move through the specimen matrix determines the mortar's durability. The amount of water that can be absorbed for a given condition is known as water absorption. In addition, it refers to hole space amounts in the specimen matrix that allows liquid mechanisms to permeate through it. The water

absorption test includes drying the cubic sample to a constant mass; the sample was submerged in water until completely saturated and the dry and saturated sample were weighted. The water absorption values were measured through the use of the following equation.

$$\text{WA\%} = \frac{W_s - W_d}{W_d} \times 100 \tag{2}$$

$W_s$ is the weight of saturated.
$W_d$ is the weight of dry.
WA is weight of absorption.

### 5.4. Water Sorptivity

The water sorptivity of a material is its ability to absorb water through the suction process. Water infiltration into the material is one of the tests connected to the material's durability. The water sorptivity of alkali-activated mortar was tested in accordance with the ASTM C1585 standard [57]. Three specimens of 25 × 25 × 25 mm were used to calculate the water sorptivity of the alkali-activated specimens. Three identical specimens for each mix were dried to a constant mass at 105 °C in an oven at the age of 28 days. The specimens were taken out and coated with silicone sealing to avoid entering the water from the sides of the specimens. After that, the specimens were kept in water with a depth smaller than the 4 mm above the bottom of the specimens as shown in Figure 4. The wet height of the specimen can be evaluated by dividing the increase in the mass of the specimen weighed at different time intervals to the bottom surface area of the specimen and density of water. These values were plotted versus the square root of time and the sorptivity index of the mortar was calculated by the slope of the best fit line.

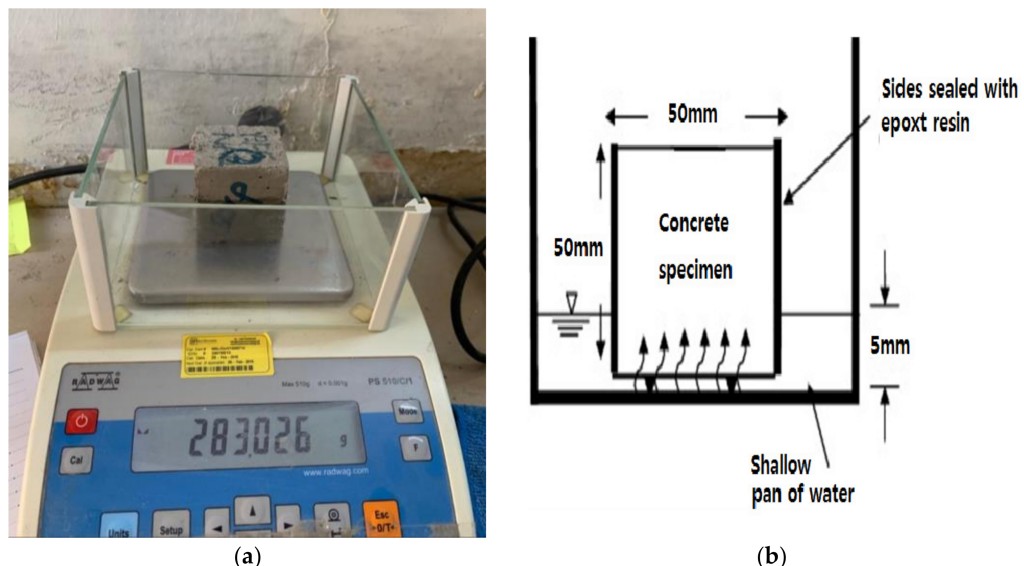

(**a**) (**b**)

**Figure 4.** The sorptivity test: (**a**) three-digit reading balance and (**b**) testing procedure.

### 5.5. Compressive Strength

The ability of a material or structure to withstand axial loads is known as compressive strength. Alkali-activated mortar cubes (50 × 50 × 50) were prepared according to ASTM C109 [58]. The mold was filled in two layers and vibrated for 30 s for each layer until the top of the specimen was leveled. Then, the molds were covered with a plastic bag for 24 h. The molds were tested at the ages of 7 days and 28 days as presented in Figure 5.

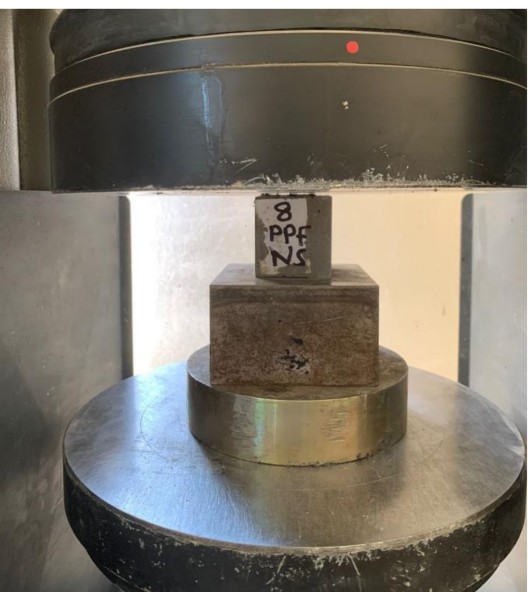

**Figure 5.** The compressive strength test of the alkali-activated mortar.

### 5.6. Flexural Strength

After 24 h of being covered with plastic and curing at room temperature, the alkali-activated mortar specimens were ready to test. Flexural tensile strength testing was performed according to British standard BS EN 196-1:2005 [59] via a 3000 kN capacity machine with a loading rate of 50 N/s (digital machine control) using a three-point bending load conducted on 40 × 40 × 160 mm identical prism specimens tested for each mixture presented in Figure 6. The flexural tensile strength was calculated using the formula below:

$$R_f = \frac{1.5 \times F_f \times l}{b^3} \tag{3}$$

where: $F_f$, $l$, and $b$ represent the peak load (N), span length (mm), and the side of the square section of the prism (mm), respectively. Figure 6 shows the details of the three-point bending test setup as well as the specimens that were tested.

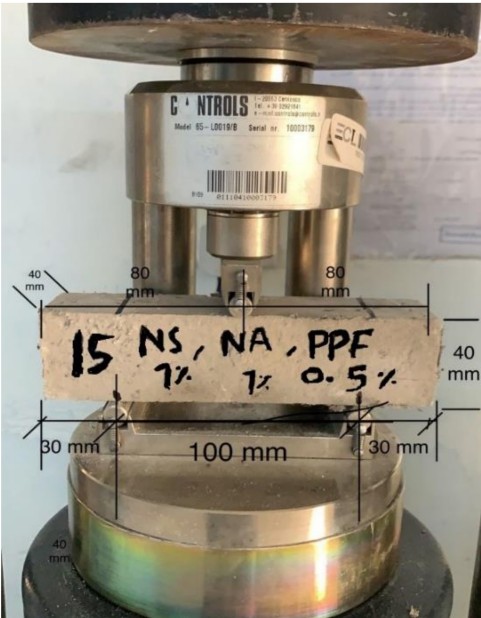

**Figure 6.** Prism specimen and basic dimensions under flexural tensile test.

## 6. Results and Discussion

### 6.1. Fresh Properties

#### 6.1.1. Flow Table Test and Setting Time

The flow table test was used according to ASTM C-230 [54] to assess the workability of the alkali-activated mortar by comparing the spread diameter of the alkali-activated mortar with the flow diameter. This experiment aimed to determine how nanomaterials and polypropylene fiber properties affected the workability of alkali-activated mortar.

Figure 7 shows how nanomaterials and polypropylene fiber (PPF) affect the workability of alkali-activated mortar. It was noted that the alkali-activated mortar without nanomaterials and polypropylene fiber (PPF) has higher workability than other mixes containing nanomaterials and/or polypropylene fiber (PPF).

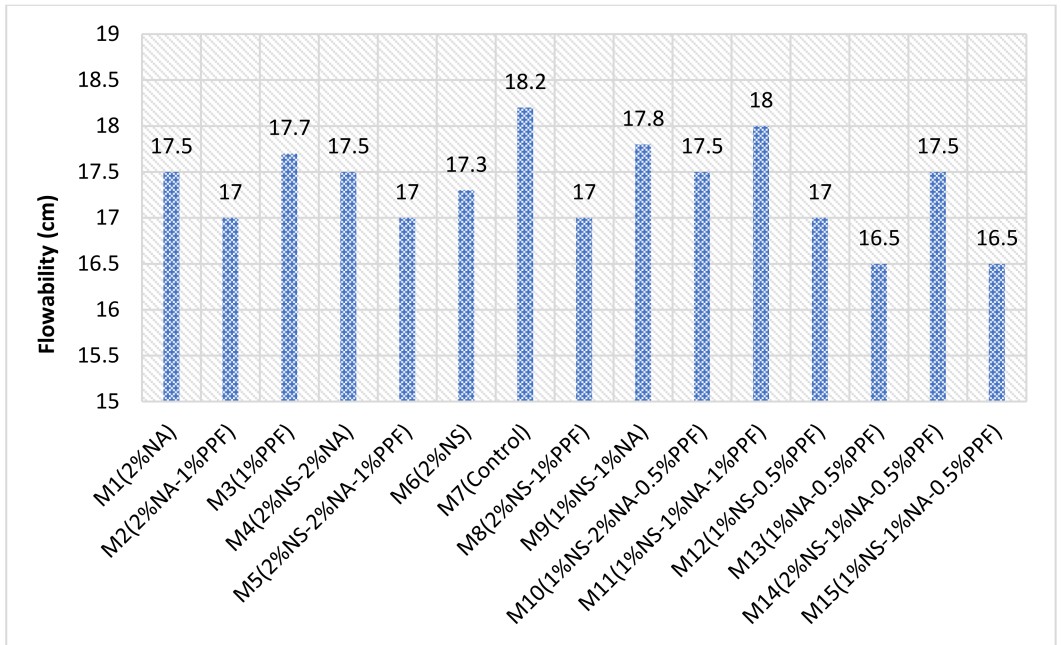

**Figure 7.** Flow table result of alkali-activated mortar (NA—nano alumina, NS—nano silica, and PPF—polypropylene fiber).

Nonetheless, samples containing 0.5% polypropylene fiber (PPF) and 1% of both nanomaterials have lower workability than samples containing 1% polypropylene fiber (PPF)-based AAM. Furthermore, it was noted that the flow ability was influenced significantly by the addition of PPF fibers. The flow ability of the mixes with PPF was very low compared with the flow of the mixes without PPF. However, the flow for all mixes was acceptable according to the classification shown in Table 5.

Moreover, the setting time of the mixes was substantially influenced by the addition of nanomaterials and PPF. The setting time of the mixes with nano silica was less than the mixes with nano alumina. The combined use of PPF with nanomaterials significantly decreased the setting time of the mixes. Meanwhile, alkali-activated mortar could be prepared under ambient conditions with the addition of nanomaterials and PPF.

#### 6.1.2. Bulk Density

An experiment was carried out to produce a bulk density of fly ash (FA)/ground granulate blasting slag (GGBS)-based alkali-activated mortar after it had been mixed. The density of each of the mixtures was calculated separately. GGBS-based alkali-activated mortar was shown in Figure 8 with its fresh unit weight density. As shown in Figure 5, the use of nano silica and nano alumina significantly affected the unit weight of AAM compared with the control mixes without nanomaterials. Whereas, the use of nano silica

or nano alumina alone decreased the unit weight values. However, the mixes with 0.5% polypropylene fiber (PPF) with both 1% of nano silica M12 and 1% nano alumina M13 conducted higher density than other mixes.

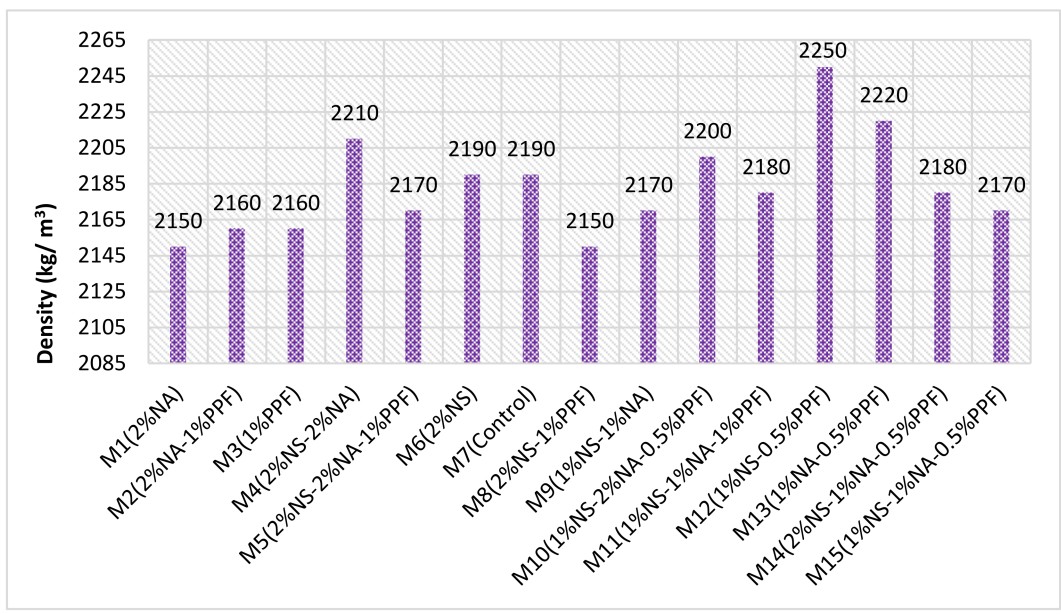

**Figure 8.** Bulk density result of alkali-activated mortar (NA—nano alumina, NS—nano silica, and PPF—polypropylene fiber).

### 6.2. Mechanical Properties

6.2.1. Compressive Strength

Alkali-activated-based fly ash (FA) and ground granulated blast slag (GGBS) involving 2% of nano silica mixing proportion was improved and higher compressive strength values were achieved [60]. In addition, shear bond strength was improved by both nanomaterials, nano alumina $Al_2O_3$ and nano silica $SiO_2$, between the alkali-activated components [61]. When compared with the control alkali-activated mortar, the compressive strength of the alkali-activated mortar with and without nano silica (NS) was significantly increased. The addition of amorphous nano silica contributes to the increase in strength [60]. Several researchers have developed various methods for producing nanomaterials-based mortar and concrete [62]. Figure 9 illustrates the compressive strength result of alkali-activated mortar with two types of nanomaterials with and without polypropylene fiber (PPF). The compressive strength was 77.20–82 MPa at 7 days and 28 days, which is the highest strength, and the mix contains 2% nano alumina. The mixtures containing 1% nano alumina (NA), nano silica (NS), and 0.5% polypropylene fiber (PPF) had the lowest compressive strength at 7 days and 28 days (46.07–55.40 MPa). When compared with mixes that contain nano $SiO_2$ and nano $Al_2O_3$ separately, the combined use of 2% for both nanomaterials negatively affected the results of the compressive strength. Whereas, the use of 2% nano silica and nano alumina alone significantly affected and improved the compressive strength of AAM. It was discovered that using polypropylene fiber (PPF) reduces compressive strength and the combined use of nanomaterials reduces the compressive strength. Meanwhile, the presence of high amounts of nanomaterials with high surface area resulted in high amounts of non-reactive parts and negatively affected the results of the compressive strength.

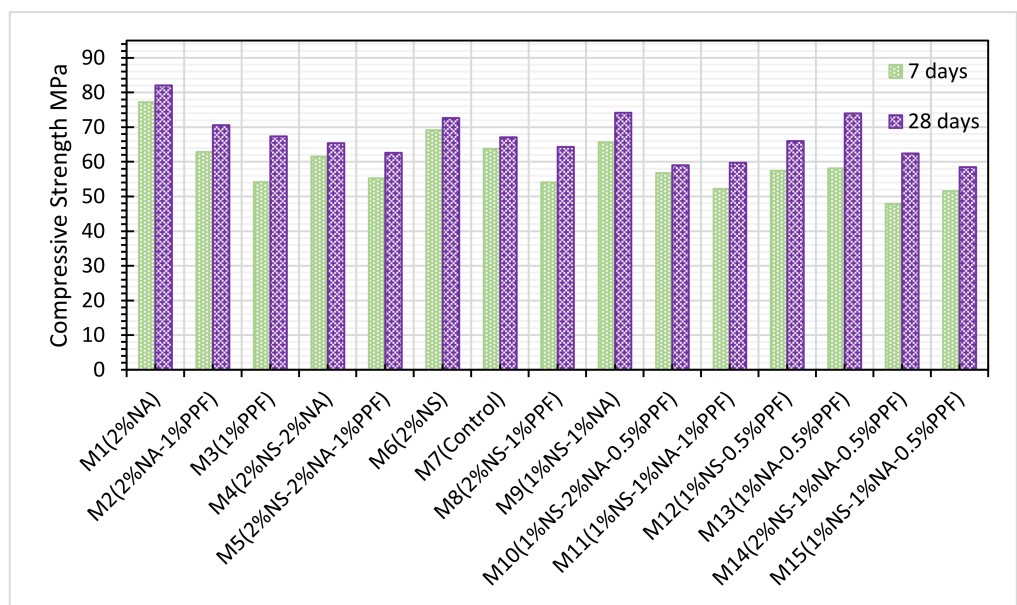

**Figure 9.** Compressive strength of alkali-activated mortar (NA—nano alumina, NS—nano silica, and PPF—polypropylene fiber).

### 6.2.2. Flexural Strength

The findings showed that alkali-resistant polypropylene fiber PPF can reduce flexural strength and improve it [63]. Figure 10 shows the flexural strength test results of alkali-activated mortar specimens after 7 days and 28 days. It can be seen that flexural strength in various mixes was reported. The mixes with polypropylene fiber (PPF) were compared with the mixes without fiber; the polypropylene fiber (PPF) has a higher efficiency in terms of flexural tensile strength and provides better performance. Fiber reinforcing is a well-known technique for improving the flexural properties and post-peak appearances of related composites by controlling fracture dissemination and spread under various types of mechanical load [64].

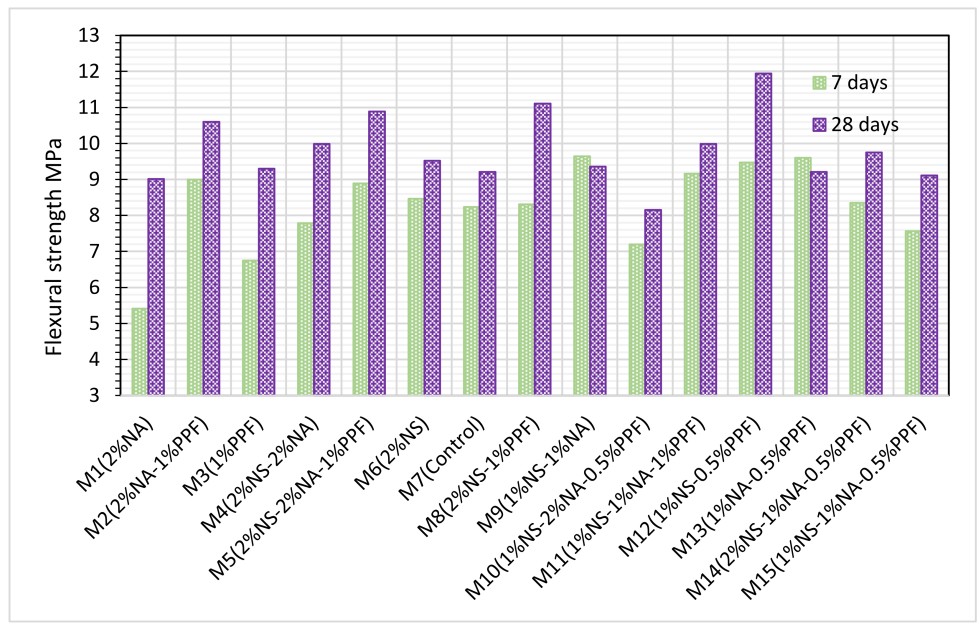

**Figure 10.** Flexural strength of alkali-activated mortar (NA—nano alumina, NS—nano silica, and PPF—polypropylene fiber).

The flexural tensile strength values were increased with time and the mixes containing 1% nano silica (NS) and 0.5% polypropylene fiber (PPF) had higher flexural tensile strength than mixes containing both 1% nano silica (NS), 1% nano alumina (NA), and 0.5% polypropylene fiber (PPF), which has lower flexural strength that is even lower than the control (without nanomaterials and PPF fibers). This finding demonstrates that the combined use of nanomaterials, particularly nano silica and nano alumina, improves the mechanical strength. In addition, AAM mixes with 1% polypropylene fiber (PPF) had lower flexural strength than the control mix at 7 days but higher at 28 days, indicating that polypropylene fiber (PPF) gives better flexural strength as a function of age. The use of 1% of nano silica and 1% of nano alumina together had flexural strength lower than the mixes containing nano silica or nano alumina alone. However, the influence of nanomaterials on the flexural tensile strength values was noted. The addition of nanomaterials significantly increased the bond strength among the alkali-activated matrix, especially the mixes with PPF fibers. The effect of nano silica on flexural tensile strength was higher than the effect of nano alumina.

### 6.3. Durability of Alkali-Activated Mortar

6.3.1. Water Sorptivity

One of the most important tests to perform is the sorptivity test when attempting to determine the capillary structure. Therefore, an experiment on water sorptivity was conducted for 28 days, as depicted in Figure 11. As illustrated in the graph, the fly ash (FA)/ground granulated blast slag (GGBS)-based alkali-activated mortar has a very low water sorptivity.

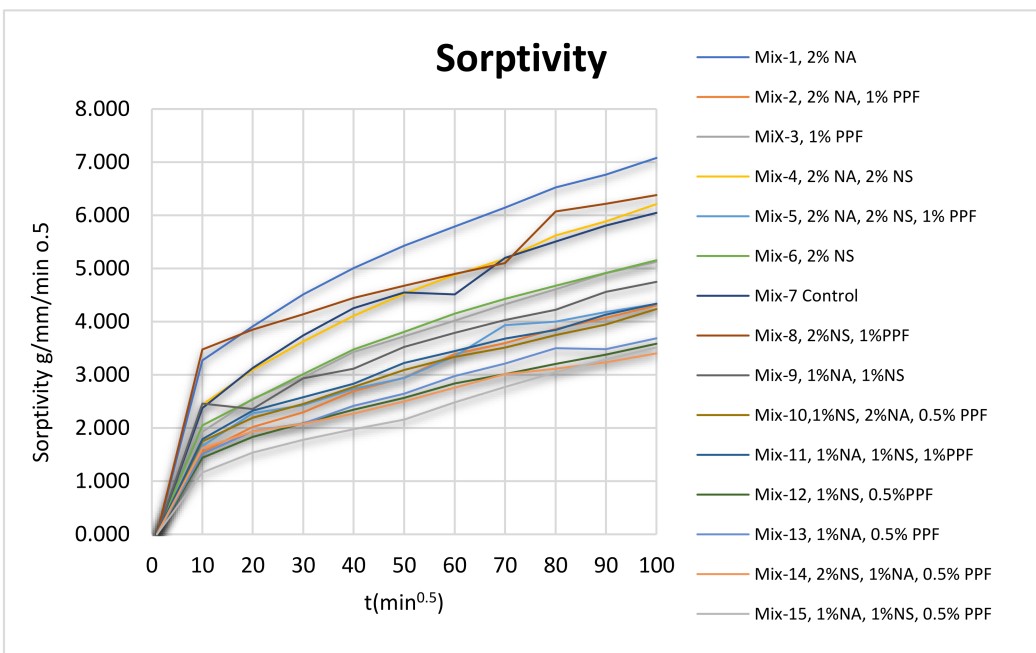

**Figure 11.** Sorptivity result of alkali-activated mortar (NA—nano alumina, NS—nano silica, and PPF—polypropylene fiber).

The samples were tested every ten minutes for the first 100 min. and the results to range around (0–3) g/mm/min$^{0.5}$ to (0–7) g/mm/min$^{0.5}$. The samples with nano $Al_2O_3$ had the highest water sorptivity compared with the samples containing 0.5% polypropylene fiber and nanoparticles. Furthermore, the structure of the matrix for the mixes containing NS and PPF was denser, which was achieved by varying the percentages of polypropylene fiber (PPF) and nanomaterial particles filling the pores. Moreover, samples containing nano $Al_2O_3$ and nano $SiO_3$ combined gave better performance than the alkali-activated mortar

containing nano $Al_2O_3$ or nano $SiO_3$ alone. The alkali-activated mortar containing 0.5% polypropylene fiber (PPF) decreased the water sorptivity and the mix containing 1% nano combined materials gave lower sorptivity than the other alkali-activated mixes.

### 6.3.2. Water Absorption

The difference in sample weight between fully saturated and oven-dried conditions is referred to as the water absorption [55]. The water absorption test was conducted on a cube (50 × 50 × 50) mm specimen after 28 days. Figure 12 shows the water absorption of a fly ash (FA)/ground granulated blast slag (GGBS)-based alkali-activated mortar as the percentage of nanomaterials and PPF changes. The mixes containing 2% of nano silica (NS) and 1% polypropylene fiber (PPF) had less water absorption—7.6% compared with 8.9% in the control. The mixes containing both nanomaterials, nano silica and nano alumina, and PPF had higher water absorption at 11.1% compared with the control, 8.9%. The porosity of the alkali-activated mortar is an important factor in its mechanical performance and durability characteristics.

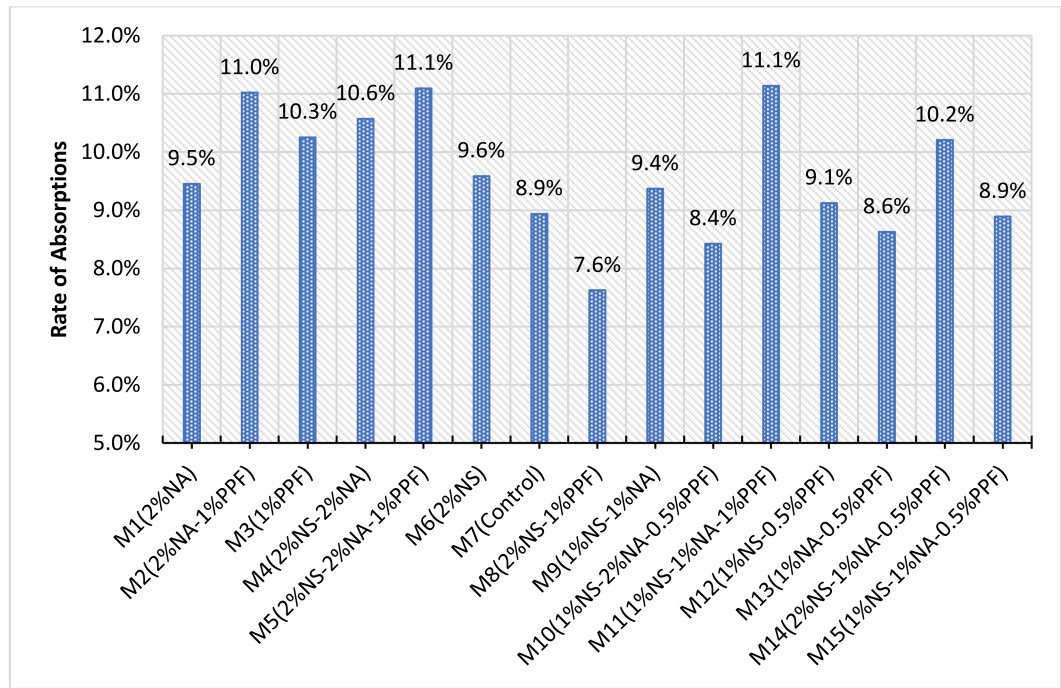

**Figure 12.** Rate of water absorption of alkali-activated mortar (NA—nano alumina, NS—nano silica, and PPF—polypropylene fiber).

### 6.4. Microstructure Analysis

#### 6.4.1. Scanning Electron Microscope (SEM)

Scanning electron microscope (SEM) photographs for the M1, M6, M9, M7, M8, M5, and M13 mixtures are shown in Figure 13. Generally, the activated samples demonstrate different microstructures depending on the types and different percentages of materials at the age of 28 days.

In samples containing 2% nano alumina, the nano alumina fillers filled porosity spaces and improved packing particles between the binder phase, FA-fly ash, and GGBFS-ground granulated blast furnace slag. The alkali-activated matrix appears to have a significant impact on the material's overall strength. Hence, this cohesion seems to improve the bond between the reactive and pore spaces, as can be observed in the SEM photos. The capillary pores increased when nano alumina was added. On the other hand, nano alumina particles appear to act as a filler in the binder structure, contributing to the formation of a denser structure by remaining in the voids. However, it may obstruct the formation of C–S–H,

which has yet to be identified. The particles in fly ash are spherical and come in a variety of sizes.

Xavier et al. [65] studied the alkali-activated mortar with the addition of NA. The NA-alkali-activated mortar presented greater quantities of alkali-activated gel and amorphous components filling the system's micro-level gaps, which could be due to the impacts of nano alumina filling the spaces to form denser alkali-activated mortars. These particles are normally hollow and have a radius of less than 10 m with smaller particles in their interior spatial structure [66]. Furthermore, the increased compressive strengths could be due to the presence of a higher amount of the glassy phase (calcium in GGBFS).

The microstructural morphology of polypropylene fiber based on alkali-activated composites are shown in Figure 13, M5, M8, and M13. It was concluded from the combined results of compressive and flexural strength that mechanical qualities in alkali-activated mortar are strongly related to micro-crack formation and polypropylene fiber to alkali-activated matrix bonding when the results of both tests are taken into consideration. Figure 13 shows the microstructure of the fiber-reinforced polymer, which is the product of the dense materials. The geo-polymerization matrix surrounds the polypropylene fiber, resulting in a high density of pores in the polymer structure. Fibers de-bonded from the alkali-activated matrix due to the polypropylene fiber's smooth surface [67].

6.4.2. X-ray Diffractions

In the XRD image for alkali-activated mortar with a 6% nano silica (NS) addition, a few peak positions were noticeable. Because of the development of a crystalline compound in the alkali-activated matrix, crystalline quartz was detected at 26–32° (2-theta) degree range [43,68]. The improved pore filling mechanisms are responsible for the improvement of the flexural tensile strength of the alkali-activated nanocomposites. When nano alumina (NA) is consistently spread throughout the matrix, it fills cavities and creates a denser microstructure [69].

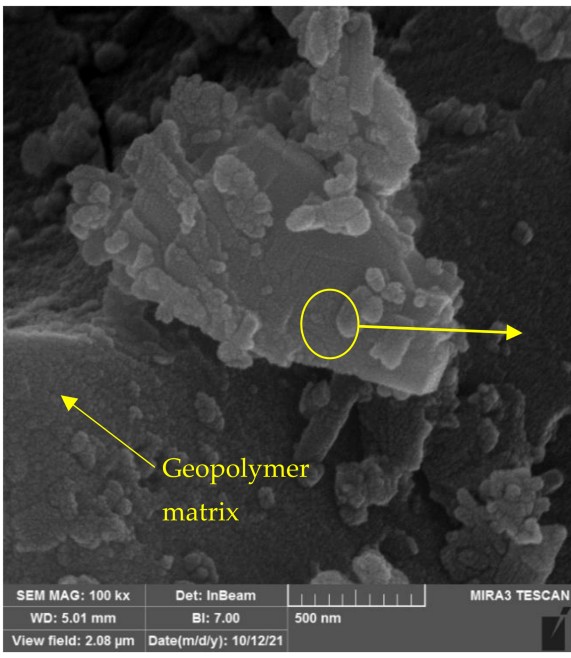 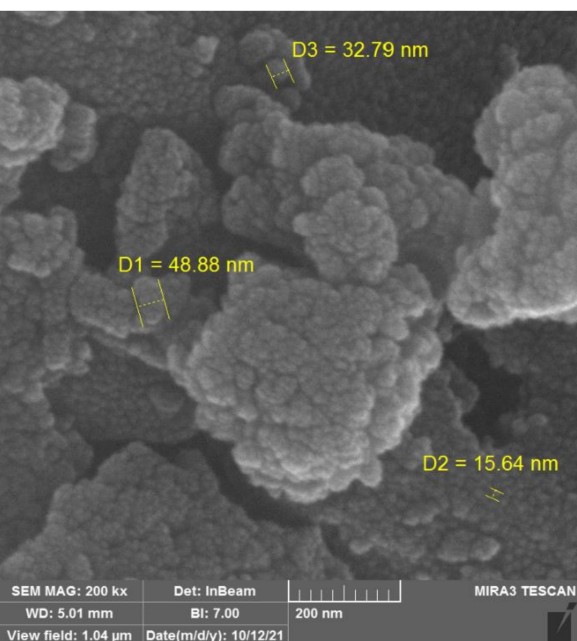

(**a**) 500 nm          (**b**) 200nm

M1, 2% of NA

**Figure 13.** *Cont.*

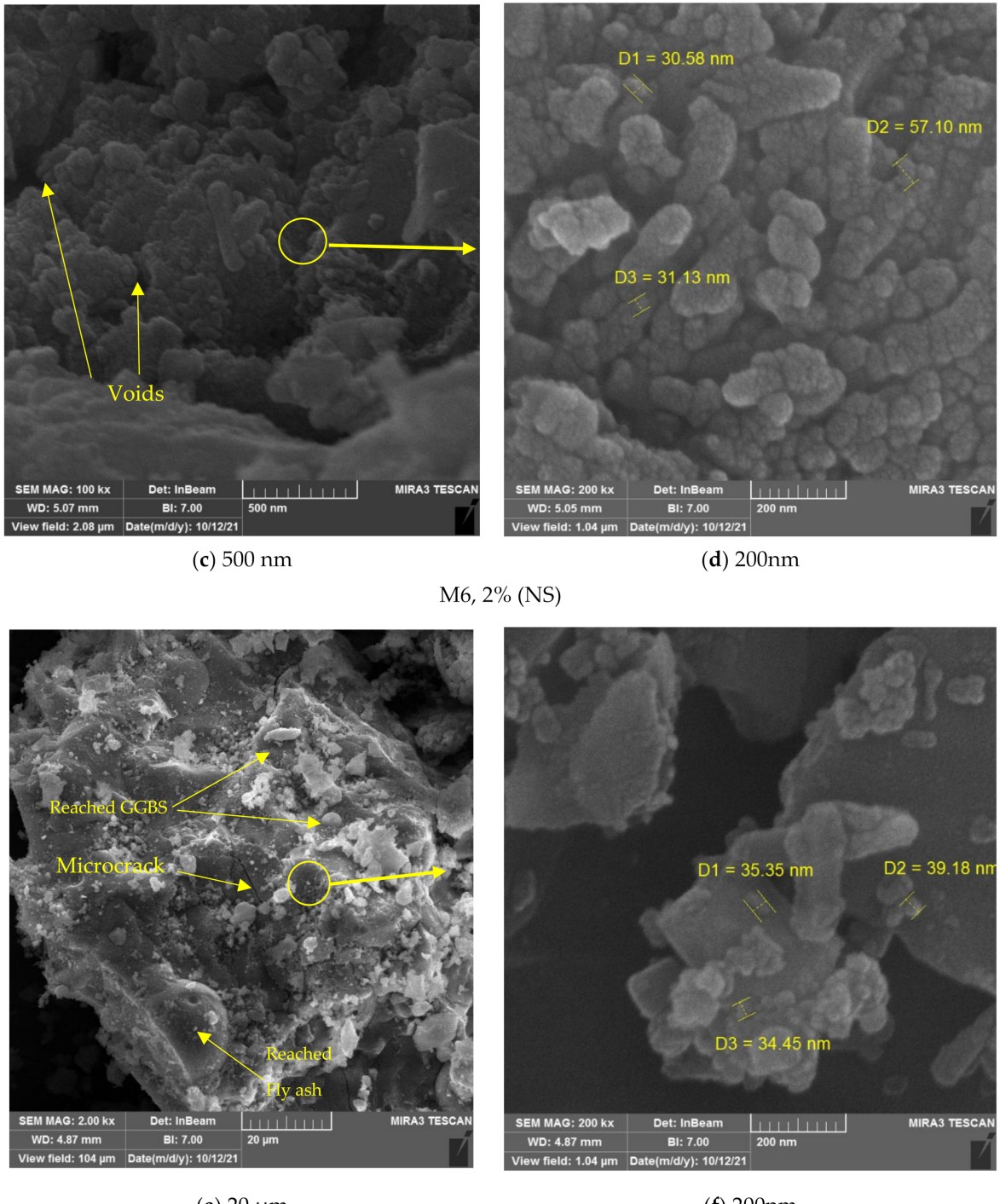

(**c**) 500 nm

(**d**) 200nm

M6, 2% (NS)

(**e**) 20 μm

(**f**) 200nm

M9, 1% of (NA) and 1% (NS)

**Figure 13.** *Cont.*

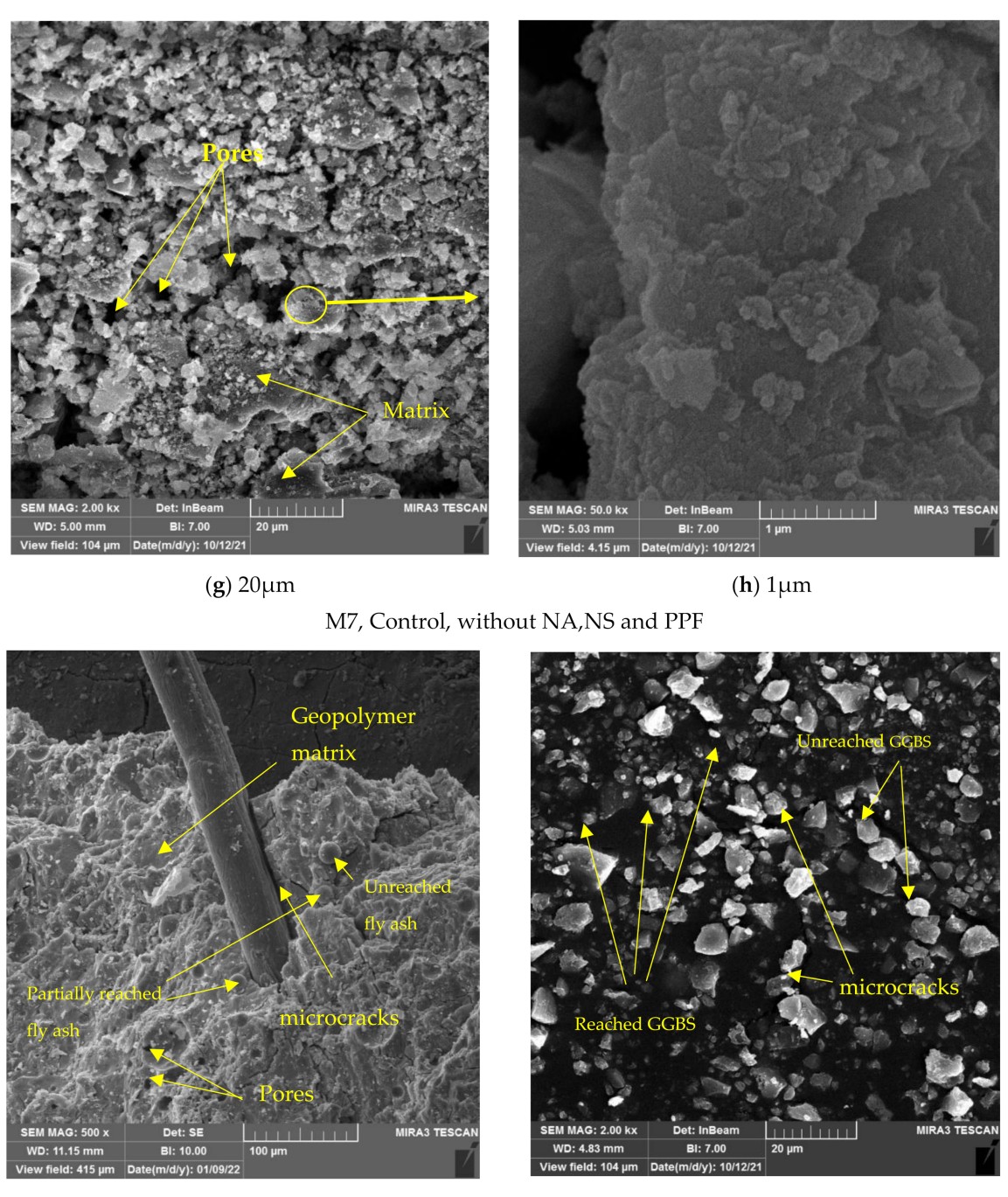

(**g**) 20μm           (**h**) 1μm

M7, Control, without NA,NS and PPF

(**i**) 100μm           (**j**) 20 μm

M8, 2% of NS and 1% of PPF

**Figure 13.** *Cont.*

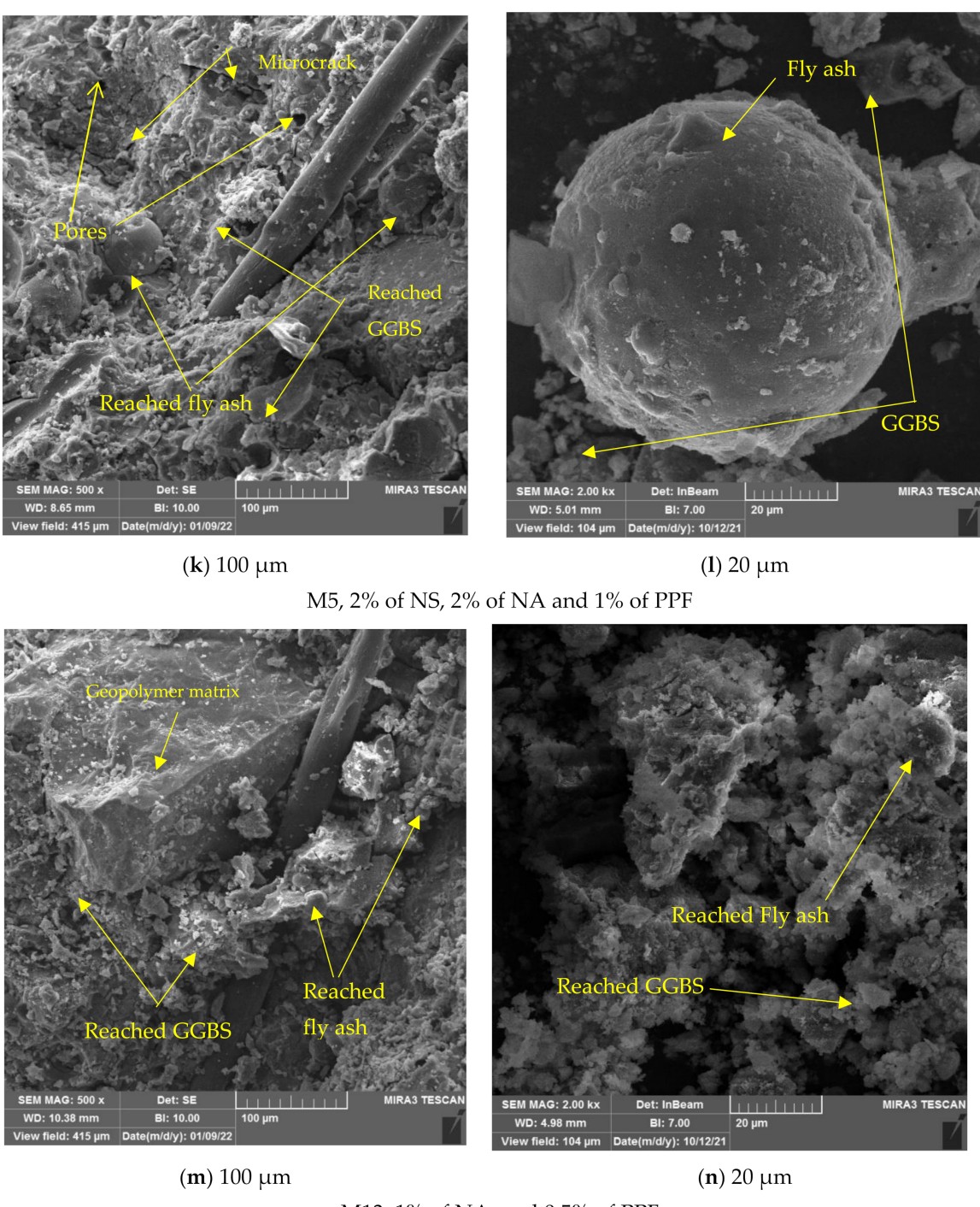

(**k**) 100 μm       (**l**) 20 μm

M5, 2% of NS, 2% of NA and 1% of PPF

(**m**) 100 μm       (**n**) 20 μm

M13, 1% of NA, and 0.5% of PPF

**Figure 13.** SEM photographs of the samples with M1, M6, M9, M7, M8, M5, and M13 (NS, NA, and PPF indicate nano silica, nano alumina, and polypropylene fiber).

X-ray diffraction (XRD) analysis produces diffraction patterns that can be used to analyze a material's structure and phases of compounds. XRD analysis was used to determine mineral phases/compounds contained in the main materials and hardened state of the alkali-activated concrete (GPC) samples [70]. The XRD patterns of nano alumina (NA), nano silica (NS), and both combinations in mix proportions of nano alumina (NA) and nano silica (NS) fly ash/ground granulated blast slag are shown in Figure 14a,b. In geopolymeric systems, nano alumina, which is composed of crystallized phases of quartz and choloalite with an amorphous hump between two values of 20–29° and an

amorphous hump between two values of 20–29°, form as a byproduct of the formation of an aluminosilicate gel. The patterns are similar when compared with those of nano silica or when they are combined, but the phases are quite different when compared with those of the alkali-activated mortar containing nano alumina or when they are combined. Alkali-activated mortar contain these filler particles because these phases were not involved in the alkali-activated reaction. Peak positions in the X-ray diffraction (XRD) analysis of alkali-activated mortar with a 2% nano silica addition were prominent in the same way as the samples containing nano silica (NS). This was due to the development of a crystalline composite in the alkali-activated mortar, which resulted in the crystallization of the quartz in the temperature range of 20–27° 2-theta degree.

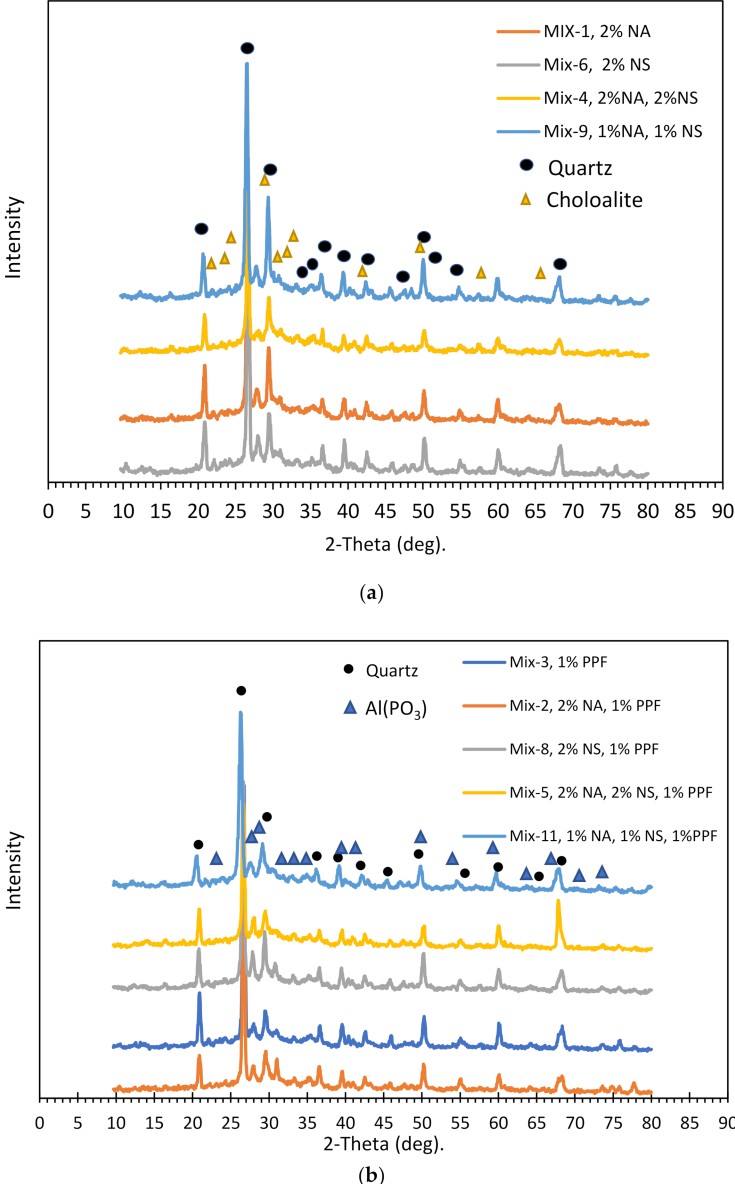

**Figure 14.** (**a**). X-ray diffraction of recent alkali-activated samples. (**b**). X-ray diffraction of recent alkali-activated samples.

XRD patterns of polypropylene fiber with and without nano alumina and nano silica were composed of the crystallized phases of quartz and aluminum metaphosphate $(Al(PO_3)_3)$ with an amorphous hump to the degree of 21–29°, which formed as a byproduct of the development of an aluminosilicate gel in alkali-activated structures. Concerning the

evaluation of the patterns, they can be explained by the XRD of the alkali-activated mortar in a recent study of samples containing polypropylene fiber which were also similar to other samples containing different percentages of nanomaterial, and the crystallization of the quartz and aluminum metaphosphate $(Al(PO_3)_3)$ phases were quite different.

## 7. Conclusions

The following conclusions were reached after examining the behaviors and the performance of nano silica (NS) and nano alumina (NA) combinations with different percentages combined with 0.5% and 1% polypropylene fiber (PPF) accumulation in a low calcium fly ash (FA) and ground granulated blast slag (GGBS) alkali-activated mortar with 12 molar concentrations cured at ambient room temperature.

1.  The addition of nanomaterials and PPF negatively affected the flow ability and workability of AAM mixes. The effect of PPF was more than the nanomaterials. The effect of NS was more than NA. The minimum workability was noted through the combined use of PPF and nanomaterials.
2.  Moreover, the setting time of the mixes was substantially influenced by the addition of nanomaterials and PPF. The setting time of the mixes with nano silica was less than the mixes with nano alumina. The combined use of PPF with nanomaterials significantly decreased the setting time of the mixes. Meanwhile, alkali-activated mortar could be prepared under ambient conditions with the addition of nanomaterials and PPF.
3.  The bulk density of alkali-activated mortar containing 0.5% PPF and 1% NS provided the highest density compared with 1% (PPF and 2% nano silica (NS)). Whereas, the mixture containing 2% nano alumina (NA) alkali-activated mortar provided the lowest density.
4.  The highest compressive strength was achieved through the use of 2% NA and the strength was (77.2–82 MPa) at the ages of 7 days and 28 days, respectively. While the lowest compressive strength was obtained through the use of 1% nano alumina (NA), 1% nano silica (NS), and 0.5% of polypropylene fiber and the strength was (51.5–58.53 MPa) at the ages of 7 and 28 days, respectively.
5.  The flexural strength of alkali-activated mortar significantly increased by the addition of polypropylene fiber and samples containing 0.5% polypropylene performed better than 1%.
6.  Water sorptivity was conducted for all the alkali-activated samples containing different percentages of materials and showed that, the alkali-activator mortar containing 2% nano alumina (NA) provided the highest water sorptivity. The samples containing both nanomaterials and 0.5% polypropylene fiber provided lower water sorptivity.
7.  Alkali-activated mortar containing 2% nano alumina (NA) and 1% polypropylene fiber (PPF) gave a high rate of water absorption. Alumina nanoparticles appear to operate as a filler in the binder structure. They may obstruct the synthesis of C–S–H because those compounds are unknown. The presence of nano alumina in the alkali-activated matrix should improve overall material strength.
8.  In the alkali-activated mortar, the development of micro-cracks and the bonding between polypropylene (PPF) and the alkali-activated matrix are both directly related to the mechanical properties of the matrix
9.  After careful examination, it can be seen that the trends in XRD patterns of alkali-activated pastes containing nano $Al_2O_3$ and both nanomaterials together are similar to those in nano $SiO_2$ and both together; the phases are extremely different. Alkali-activated mortar contains these filler particles because these phases were not involved in the alkali-activated reaction. The samples containing polypropylene fiber were also similar to other samples containing different percentages of nanomaterials, and the crystallization of quartz and Al $(PO_3)$ phases were quite different.

## 8. Recommendation and Future Perspective

The following recommendations are provided to researchers and practitioners in this field. A critical further investigation of the different types of fibers with different types of nanomaterials as well as investigation of waste tile texture is strongly advised. In addition, utilization of various types of recycled materials as fine aggregates in alkali-activated mortar is recommended to improve the mechanical and durability performances of alkali-activated mortar. In the future, the mechanical properties of alkali-activated mortar under chemical resistance of alkali-activated mortar, as well as length change tests of alkali-activated mortar containing the same materials under different conditions, will be investigated.

**Author Contributions:** Conceptualization, M.H.D. and M.A.M.; methodology, M.H.D.; software, M.H.D., M.A.M. and R.A. validation, M.H.D.; formal analysis, M.H.D. and M.A.M. investigation, M.H.D.; resources, M.H.D. and R.A. data curation, M.H.D.; writing—original draft preparation, M.H.D., M.A.M. and R.A. writing—review and editing, M.H.D. and R.A; visualization, M.A.M.; supervision, M.H.D. and R.A.; project administration, M.H.D.; funding acquisition, M.A.M. All authors have read and agreed to the published version of the manuscript."

**Funding:** This research received no external funding.

**Institutional Review Board Statement:** Not applicable.

**Informed Consent Statement:** Not applicable.

**Data Availability Statement:** Generated during the experimental study.

**Acknowledgments:** Kindly, I would like to express thanks and my sincere gratitude to my research supervisor, Assistant professor Mohammed Ali, for continuous support during my Ph.D., and for his patience and motivation; and Radhwan Alzeebaree for contributing his abilities in the experimental study and providing guidance in the collection of the main materials and experimental working environment.

**Conflicts of Interest:** The authors declare no conflict of interest.

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
