# Peer review of "Performance of Fiber-Reinforced Alkali-Activated Mortar with/without Nano Silica and Nano Alumina"

_sustainability, doi:10.3390/su14052527_

Round 1

Reviewer 1 Report

In Figures 8 and 10, the authors use “mix-1”, ”mix-2”, etc to indicate types of mortars, which can be referred to Table 3. However, the authors also use abbreviated names for different types of mortars in figures 4 to 9 (such as GP-NA, GP-NA2-PP1) without any explanation. Please provide some explanation for the name.

Author Response

Kindly, I went over your all valuable comments.

Reviewer 2 Report

The first time abbreviations appear, they must come together with the word in question. Review the entire document and correct it.

Introduction:
The use of nanosilica and nano alumina is not duly justified, since the advantages indicated in line 53 refer to zinc oxide. Please, duly justify the use of these nanomaterials and indicate their disadvantages, which also have them. As is its caking due to its high specific surface area and the need to use superplasticizers for this. They are used, although their use is not justified throughout the document.
In paragraph (64-68) you indicate a high resistance to compression with the use of fly ash, but the reference that you name indicates the opposite in Figure 7, which even decreases as the fly ash content increases. Please put that paragraph in context or if you do not look for another reference that does demonstrate what you indicate.
‘’ The design-expert software creates varied designs by adding variables to the programs and running the program. ’’ This program that you mention in the abstract is not developed in the introduction either. What program? The program used in reference 27 is a statistical program with a different experimental design since it does not introduce nanomaterials, GGBS or alkali activated. Please justify the use of that software properly.

You must also indicate where you get the percentage of 50% reduction with FA that you indicate on line 92.
I believe, in general, that the introduction does not adequately justify the materials and methods used in the experimental phase. Why do you use Nanosilica and nanoalumina? Why do you use alkali activated? Why FA and GGBS and not other geopolymers? Is it not necessary to use specific additives? It seems to me an essential element to only indicate that 1 and 2% of additives are used without even indicating which one. Fiber reinforced are not even mentioned in the introduction, but they are included in the title.

Materials and methods
On line 138 you justify the percentage of 2.5 to economic considerations. What considerations?
The program used in reference 27 is a statistical program with a different experimental design since it does not introduce nanomaterials, GGBS or alkali activated. Please justify the use of that software properly.
The statements made about high compressive strengths compared to references 22 and 32 are not entirely acceptable. Reference 32 doesn't even have compressed results in its article. Reference 22 only has it with fly ash and although it is much lower than your results, I believe that it is not properly justified with just a reference.

Despite the comments expressed above, the topic of the research seems to be interesting for the research community. I would be glad to see the paper appear in print provided that major improvements are made.

Author Response

(The authors gave the same response as above.)

Reviewer 3 Report

The article deals with an interesting topic that has been widely developed in the last decade. Unfortunately, the introduction lacks some important literature items that have been published in recent years. The authors test their samples after 7 and 28 days, while usually samples are tested after 7, 14, 21 and 28 days to determine the behavior of the material (including geopolymer). Sometimes even, if the process of increasing of the mechanical parameters does not stabilize after 28 days, samples are also tested after 60 or even 90 days. There is an extensive literature available on this topic. Besides, the authors do not state how many samples were tested for each batch for each type of mixture. If the test results are to be of any value, they must be statistically processed. Single results are always unreliable, so it is not possible to draw quantitative conclusions. Moreover, the article requires, in many places, broader comments and supplements. Contains typos and grammar errors. Should be checked by a native speaker.

General remark. Before using each abbreviation in the text, it is necessary to specify its full development, e.g. Ordinary Portland Cement (OPC), Nano alumina (NA), Nano-silica (NS), etc. Therefore, using only abbreviations in the summary is unacceptable in this case. The abstract should be redrafted to make it understandable for the potential reader. Some detailed comments below.

  1. Page 1. Line # 34. The abbreviation OPC was used for the first time without explanation. It should be corrected.
  2. Page 1. Line # 36. Difference in the use of the abbreviation granulated blast furnace slag. Sometimes the authors use the GGBS abbreviation and other times GGFBS. The fact that the article was written by several authors does not exempt from the necessity to carefully correct it and standardize the entries.
  3. Page 1. Line # 41. GPC abbreviation used without explanation.
  4. Page 2. Line # 48. The abbreviation FA was used for the first time, while the term fly ash had been used before, and without its abbreviation. It should be corrected.
  5. Page 2. Line # 57. Authors wrote, “A geopolymer is a good substitute for regular concrete”. Such a firm statement is not true. There are many studies available that clearly show the limitations of the use of geopolymers, especially modern geopolymers with various inorganic additives. The fact that the Romans used a certain type of geopolymer (the so-called Roman cement) does not entitle us to say as above.
  6. Page 2. Line # 84. The abbreviation GPM was used without explanation.
  7. Page 2. Line # 92. GBFS8 abbreviation used without explanation
  8. Page 2. Lines # 98-99. The abbreviations FA-BGPC and SS / SH were used without explanation.
  9. Page 3. Line # 112. The American standard ASTM C 618 is not cited in references. It should be corrected.
  10. Page 4. Line # 134. The information "... demonstrated in figure 1,2" is incomprehensible.
  11. Page 4. Table 3. All quantities appearing in this table should be described (defined). Only abbreviations cannot be given in the table column headings. As it stands, this table is largely unreadable.
  12. Page 5. Lines # 147-152. What were the curing conditions? It is only insufficient to say that in the "laboratory environment". The temperature and humidity should be specified. Besides, why the samples were only stored for 7 and 28 days. Typically, samples are tested after 7, 14, 21 and 28 days to determine the behavior of the material, including the geopolymer. The conditions adopted by the authors require a broader explanation. Additionally, it is necessary to inform how many samples of each type of mixture (m1, M2, etc.) have been tested in a given series. This part of the research description should be completed.
  13. Page 6. Line # 167. The American standard ASTM C 138 is not cited in references. It should be corrected.
  14. Page 6. Equation (2) requires a description of all values.
  15. Page 7. Line # 210. The American standard ASTM C 109 is not cited in references. It should be corrected.
  16. Page 8. Line # 220. The RILEM 50-FMC / 198 Committee document is not cited in references. It should be corrected.
  17. Page 8. Line # 225. Wrong number of figure. It should be Figure 5. Besides, there are dimensions wrongly marked in this drawing. The distance between the supports should be 100mm. The distance from the end of the beam to the edge of the support, (steel roller) is unnecessary. What does the 50mm dimension refer to in this drawing?
  18. Page 8. Line # 232. The American standard ASTM C 124 is not cited in references. It should be corrected.
  19. Page 9. Fig.4. The method of marking the tested mortar mixtures should be explained. Why are there different designations of mixtures in Table 3 than in this and the next figure?
  20. Page 10. Fig.6. The graphs in this figure show little. Only how the compressive strength of individual mixtures changed after 7 and 28 days. On the basis of the presented research, it is impossible to determine the nature of changes in strength as a function of curing time. Was the strength increasing process the same, similar, or different for the varying blend compositions? For this, it is required to get additional results after 14 and 21 days. If there are none, these results are incomplete and of little interest to the potential reader.
  21. Page 11. Fig.7. Notes as for Fig. 6. Moreover, as the authors explain, the flexural strengths after 7 and 28 days for the GP-NS1-NA1 and GP-NA1-PP0.5 blends are the same while the compressive strengths (Fig. 6) are different. This requires in-depth discussion and comment.
  22. Page 13. Figs.10a and 10b. Why the markings of the mixtures are different than in the previous drawings? Why in these figures only show selected mixtures and not all of them? This is incomprehensible and requires a broader comment.
  23. Page 14. Lines # 358-362. This sentence is incomprehensible. It needs to be corrected.
  24. Pages 14-15. Conclusions. Requests should be redrafted. Some are obvious and others contain only detailed research results. The last conclusion is disputable.

Summarizing, in my opinion the scientific level of this article is low. If the authors will not sufficiently complete the research data and analysis of the results, or if only single samples of a given mixture were tested, I advise you to reject them.

Author Response

(The authors gave the same response as above.)

Reviewer 4 Report

The manuscript needs to be sent for proofing before it can be accepted for the second review. It is not well written! There are some hanging or incorrect or misleading sentences in the manuscript. Below are some examples.

Line 32 to 35: Geopolymer obtains compressive strength by containing a high range of silica and alumina and a high alkali content, unlike OPC or pozzolanic cement [2]. Increasing percentages of binders such as pozzolanic waste products, fly ash, rice husk ash, and pulverised granulated blast furnace slags (GGBS).

Line 48 to 49: Evaluations assessment provides a complete geopolymer concrete's physical properties, mechanical properties, microstructure, and mix design collected from diverse sources [7][8].

In addition, the authors need to address a few concerns of the manuscript before it can be reconsidered for publication in the journal.

Line 31: What is the alkaline condition for the geopolymerization to happen? What are the chemicals involved for the alkaline activation? Specify clearly what triggers the alkaline activation of geopolymer!

Line 42 to 43: It is wrong to state that geopolymer is frequently made from industrial by-products like fly ash and slag. Alkaline activators in the form of sodium hydroxide and sodium silicate are the necessary ingredients of geopolymer. Why is there no review on other common precursors for geopolymerization such as rice husk ash and silica fume? 

Line 51: What is the standard referred to for conducting the flow table test?

Line 140: What type of superplasticizer was used for the geopolymerized material?

Line 301: How durable is the geopolymer mortar against abrasion, acid attack, heat, and chloride ion penetration? Assessment of the durability performance of the geopolymer mortar under the study was limited to the water sorptivity and water absorption tests.

Line 332: The microstructural analysis of the geopolymer mortar should also include Scanning Electron Microscopic results. Please provide and discuss in detail images of the Scanning Electron Microscopic images of the geopolymer mortar under the study.

Author Response

(The authors gave the same response as above.)

Reviewer 5 Report

The topic of the paper is interesting but the paper has not been structured properly. The novelty of the paper is missing. The following are required to be considered by the authors before the paper being accepted. 

  • In the Abstract, you need to define the NA, NS, FA, GGBS, and NaOH
  • The gap in the literature has not been stated and the novelty of the current research is missing. Therefore, these must be addressed in the revised version.
  • Where did the authors get the information in Table 5?
  • The current conclusions are massive, and it contains lots of unnecessary information. Therefore, authors are firstly required to state the main aim of the current research at the conclusion before listing the main findings. Then the findings should be summarised in and shortened as they are seems to be too much information there.

Author Response

(The authors gave the same response as above.)

Round 2

Reviewer 2 Report

The article has improved a lot compared to the first version, the introduction has been completely rewritten and although much necessary information has been added, but it is still difficult to understand. I recommend a grammar check by a native speaker before it can be accepted.

On the other hand, the data from the Flow table test are very important and it appreciate their incorporation, but they also leave evidence of the existence of a variance from 18.2 to 16.5 that is worrying since it can greatly distort the data. In addition, since the results are only available at 7 and 28 days, it is even more difficult to determine the behavior of the material. There is an extensive literature available on this topic. Single results are always unreliable, so it is not possible to draw quantitative conclusions.

Author Response

Kindly, the all comments have responded. 

Reviewer 3 Report

After the authors introduced very extensive corrections and additions, the article gained much in correctness and scientific quality. Unfortunately, there is one other issue that has not been well corrected, which is

1. Page 6. Fig.2 and Page 9. Line #297. There is a difference between the view of the samples for determining the compressive strength shown in Fig. 2 (all samples are cuboid) and the dimensions of these specimens given on page 9, i.e. (50x50x50mm), i.e. cubic samples. This requires some explanation.

2. Page 9. Lines # 306-314 and Fig.6. If the trabecula was tested as shown in Fig. 6, then formula (2) should be as follows:

fflex = 3(Pmax • L)/2(b •d2)

Pmax - peak load (N);

L = 100 mm;

b = 40 mm;

d = 40 mm;

a = 0 mm

because as it is seen in Fig. 6, the beam did not have a notch, therefore, notch depth a = 0.

L is the span length of the specimen; therefore, Fig.6 needs to be corrected, because it is the distance between the support points, while in Fig. 6 this quantity is absent. Only the distances from the ends of the sample to the edges of the steel rollers (30 mm each) and the distances from the axis of the sample to the unknown point, because the drawn arrows cover them, were marked. The distance between the supported rollers, as well as the total length of the specimen should be clearly marked. The rest can be easily calculated. Therefore, Fig. 6 needs to be corrected to be legible.

Second, why was the procedure described in the 50-FMC Draft Recommendation followed? It concerns the determination of fracture Energy by means three-point bend test on notched beams. The authors do not use beams witch notches and do not set fracture energy. It would be more correct to base e.g. EN 196-1: 2005. Methods of testing cement - Part 1: Determination of strength, CEN, February 2005.

Author Response

kindly, all comments have responded.

Reviewer 4 Report

The manuscript has a lot of grammatical mistakes. The authors need to send the manuscript for proofreading by English experts before it can be resubmitted for review again. At its present state, the manuscript is unacceptable!

Author Response

kindly. all comments have responded.

Round 3

Reviewer 3 Report

The article still requires some minor corrections and careful checking for typing errors.

Noticed corrections necessary to make in the text:
1. Page 4. Line # 169. Delete the full stop at the beginning of the sentence.
2. Page 8. Line # 276. Replace "}" with "]".
3. Page 9. Line # 303. In the Ff notation, the index "f" should be subscript.
5. Page 10. Fig.6 (b). The description of this drawing should, in my opinion, be as follows: Fig.6. View of the: (a) ready to test beam specimen and basic dimensions, (b) scheme of the determination of flexural strength acc. to [58].
6. Page 25. Line # 690. Incorrect description of reference [58]. It should read: BS EN 196-1: 2005. Methods of testing cement - Part 1: Determination of strength, BSI, London, 2005.

Author Response

kindly, I went over all comments and have corrected them and replaced them.

Reviewer 4 Report

The manuscript can be accepted.

Author Response

Thanks for your acceptance.